# Chronic Breathlessness in Obstructive Sleep Apnea and the Use of Lymphocyte Parameters to Identify Overlap Syndrome among Patients

**DOI:** 10.3390/jcm12030936

**Published:** 2023-01-25

**Authors:** Dan Liu, Zhiding Wang, Yan Zhuang, Yan Wang, Jing Zhang, Rui Wang, Jie Cao, Jing Feng

**Affiliations:** 1Department of Respiratory and Critical Care Medicine, Tianjin Medical University General Hospital, Tianjin 300052, China; 2Beijing Institute of Basic Medical Sciences, Beijing 100850, China; 3Department of Respiratory Medicine, Tianjin Medical University Second Hospital, Tianjin 300052, China

**Keywords:** OSA, breathlessness, OVS, lymphocytes, intermittent hypoxia

## Abstract

Little is known about the distribution of etiology in obstructive sleep apnea (OSA) combined with chronic breathlessness. A significant portion of patients in this group have so-called “overlap syndrome (OVS)”, characterized by chronic obstructive pulmonary disease (COPD). OVS has more complications and a poorer prognosis compared to patients with either OSA or COPD alone, which makes it important to identify OVS early in OSA. The current study was a retrospective cross-sectional analysis of consecutive adult patients who were diagnosed with OSA (*n* = 1062), of whom 275 were hospitalized due to chronic breathlessness. Respiratory and cardiac diseases accounted for the vast majority of causes, followed by gastrointestinal and renal disorders. The final study population comprised 115 patients with OSA alone (*n* = 64) and OVS (*n* = 51), who had chronic breathlessness as the primary complaint, not secondary as one of many other complaints. Lymphocytes, CD4 counts, neutrophil-to-lymphocyte ratio (NLR), and PLR were differently expressed between the OSA-alone group and OVS group. The NLR, lymphocytes, and CD4 counts had a moderate diagnostic value for OVS in OSA patients, with AUCs of 0.708 (95% CI, 0.614–0.802), 0.719 (95% CI, 0.624–0.813), and 0.744 (95% CI, 0.653–0.834), respectively. The NLR had the highest AUC for predicting a 6-month re-admission of OVS, with a cut-off of 3.567 and a moderate prognostic value. The sensitivity and specificity were 0.8 and 0.732, respectively. In the animal model, the spleen hematoxylin- and eosin-stained, electron microscopy images showed germinal-center damage, chromatin activation, and mitochondrial swelling under the overlapping effect of intermittent hypoxia and cigarette smoke exposure. OSA with chronic breathlessness cannot be overstated. A significant proportion of patients with COPD in this group had poor lung function at initial diagnosis. The NLR is a useful biomarker to differentiate OVS among OSA patients combined with chronic breathlessness.

## 1. Introduction

Obstructive sleep apnea (OSA) is a clinical condition of sleep disorder, with the hallmark of recurrent partial-to-complete upper airway collapse during sleep, leading to hypercapnia and repetitive arousal and intermittent hypoxia [1,2]. OSA affects close to one billion adults aged between 30 and 69 years [3]. The implications of the high prevalence of OSA are immense, particularly in relation to the occurrence and progression of chronic diseases [4,5]. Our team has previously confirmed the decline of cognitive function in an OSA rat model [6]. Many clinical studies have confirmed that OSA leads to cognitive dysfunction, such as deficits in attention, memory, and executive ability. Patients with OSA often have nocturnal awakenings, and some patients have symptoms of chronic breathlessness due to complications associated with OSA, such as cardiovascular and chronic lung disease. OSA is increasingly valued by multiple disciplines, not only in respiratory medicine. From a dental and orthodontic point of view, clinicians have also gradually increased their awareness of this disease [7,8], and many new treatments have emerged in recent years [9]. Chronic breathlessness is a distinct clinical syndrome, and it is associated with poor physical and mental health-related quality of life [10]. The detection of chronic breathlessness is important because it is associated with a two-fold greater increase in cardiovascular and all-cause mortality over 10 or more years [11]. However, data are lacking on the study of chronic breathlessness in OSA. Clinically, we often see OSA patients delay seeing a doctor because they overattribute their symptoms to obesity and age problems, although these have been noted to cause breathlessness in some studies [12,13,14]. Due to cognitive dysfunction caused by OSA, it is also easy for patients to lack awareness of their own diseases and delay medical attention. Moreover, many patients are unable to accurately describe their corresponding symptoms, which can result in referrals to the wrong department.

Chronic obstructive pulmonary disease (COPD) is one of the most common diseases that cause decreased lung function and chronic breathlessness in OSA. A significant number of patients with OSA have so-called “overlap syndrome” (OVS), characterized by COPD [15]. Due to the additive respiratory pathophysiology of OSA and COPD, patients with OVS have a greater decrease in oxygen saturation [16] and an increased risk of developing cor pulmonale [17] relative to patients with either OSA or COPD alone.OSA and COPD both show changes in various immune cells, particularly in lymphocytes [18]. The inflammatory process induces an increase in neutrophils and platelet count, accompanied by a decrease in the lymphocyte count, making immune-cell ratios valuable biomarkers for evaluating both inflammatory status and cell-mediated immunity. The neutrophil-to-lymphocyte ratio (NLR) and the platelet-to-lymphocyte ratio (PLR) are two extensively used inflammatory markers for evaluating inflammation in several diseases [19,20]. OVS has the potential to cause a more intense inflammatory process than either OSA or COPD alone. Thus, we aimed to explore the potential role of various lymphocyte parameters to identify OVS among OSA patients with complaints of chronic breathlessness.

In this cross-sectional study, we systematically reviewed patients diagnosed with OSA and with the symptom of chronic breathlessness at our sleep center and classified their etiology. Our aim was to increase clinician attention and to derive the distribution of etiology in OSA patients combined with chronic breathlessness. COPD is one of the most common chronic lung diseases associated with OSA, and many previous studies have confirmed that OVS has a worse prognosis than either COPD or OSA alone. Previous studies have focused on screening patients with COPD for OSA; however, we found that many patients with OSA have a delayed diagnosis of COPD. Clinically, many OSA patients are obese, mood-disordered, and rarely move and communicate with other people. Some patients with OVS will gradually experience hypercapnia and even respiratory failure. Compared with early drowsiness caused by simple OSA, the lethargy and fatigue caused by these conditions are easily ignored by patients. Through face-to-face consultation, we can easily identify patients with suspected OSA according to their obesity, anatomical abnormalities, and typical symptoms, such as drowsiness and fatigue. Thus, we hope to improve the early diagnosis and prognosis of these COPD patients through clinically accessible biomarkers, which will help the clinician to triage patients more efficiently and accurately. Through an animal model, we further investigated the damage to the immune system in OVS compared to OSA or COPD alone. We also attempted to provide new clues for the treatment of OVS.

## 2. Materials and Methods

### 2.1. Study Design and Participants

The present investigation was a retrospective cross-sectional study involving consecutive adults who underwent an in-laboratory sleep recording and were diagnosed with OSA from January 2018 to October 2021 at the Sleep Medical Center, Tianjin Medical University General Hospital (Heping, China). First, we obtained the distribution of causes in OSA patients with chronic breathlessness. Study participants were patients with the complaint of chronic breathlessness, defined as breathlessness of at least 8 weeks in duration, without an acute exacerbation in the 4 weeks before assessment. Patients who had been recurrently hospitalized for the same cause of breathlessness were excluded. This group of patients was called, Entire OSA Cohort. Further, we aimed to investigate early recognition in patients with COPD. During the first visit, the patient may have had other major complaints that helped to determine the condition for triaging to the appropriate section. Thus, we only included patients with chronic breathlessness as the primary complaint, not secondary, as one of many other complaints. Patients with lung diseases that needed to be differentiated from COPD, such as interstitial lung disease (ILD), asthma, or lung cancer, were also excluded. This group of patients was selected from Entire OSA Cohort and defined as Final Study Population. We also recruited healthy volunteers as controls. They underwent sleep monitoring as well as spirometry. People who were without OSA or COPD were finally recruited as healthy controls.

All data were anonymous and complied with the requirements of authorities for personal data protection. This study protocol was approved by the Ethical Committee of Tianjin Medical University General Hospital (IRB2019-WZ-175), and informed consent was obtained from each participant. To ensure data accuracy, we included only hospitalized patients and conducted rigorous screening for enrollment.

### 2.2. Clinical Data and Laboratory Tests

We diagnosed the cause of chronic breathlessness based on accepted diagnostic criteria [21]. The demographic characteristics of all of the subjects were collected. Breathlessness was assessed using the modified Medical Research Council (mMRC) breathlessness scale [12,22]. The mMRC is a five-point ordinal scale (0–4) correlating to the level of exertion before it is limited by breathlessness. Higher scores reflect higher functional impairment due to breathlessness. Breathlessness was defined as an mMRC score of ≥ 2 for the current analysis. COPD severity, including airflow-limitation severity and combined COPD assessment, were evaluated by investigators according to the GOLD 2017 criteria [22]. The symptomatic assessment was based on both the COPD Assessment Test (CAT) and the mMRC scores. Participants had no history of hospital admissions with COPD exacerbations. COPD was graded into four stages on the basis of FEV1: grade 1 (FEV1 ≥ 80), mild COPD; grade 2: (FEV1 50–79), moderate COPD; grade 3: FEV1 30–49%, severe COPD; and grade 4: FEV1 < 30% or FEV1 30–50%, very severe COPD.

The venous blood samples were collected early in the morning before breakfast. Analysis of the blood-lymphocyte subtypes was performed using flow cytometry. Fresh blood was stained with anti-CD3 (total T lymphocytes), anti-CD45 (total leukocytes), anti-CD4, and anti-CD8 (all from Becton Dickinson (BD), San Jose, CA, USA) [23]. All procedures were performed according to the manufacturer’s instructions. The samples were analyzed using FACS Calibur (BD Pharmingen, San Jose, CA, USA) and FlowJo software Version 10 (Tree Star, Ashland, OR, USA). The white blood counts were measured using an automatic hemocytometer. The total cell number of a particular lymphocyte subpopulation was calculated from the WBC and then the frequency of a given population in flow-cytometry analysis was reported as cells/μL.

### 2.3. Sleep Monitoring

All participants underwent a single full night of sleep recording at the Sleep Medical Center. They were permitted to follow their habitual sleep time from 21:00–22:00 h to 06:00–07:00 h. Each patient’s sleep was monitored using polysomnography (PSG) (Alice 5; Philips Respironics, Murrysville, PA, USA) or a portable sleep-monitoring recorder (Alice PDX; Philips Respironics, Murrysville, PA, USA), with continuous monitoring by two technicians. Sleep parameters were scored manually using the American Academy of Sleep Medicine (AASM) Manual v2.3 2016 [24].

Respiratory sleep patterns were studied according to AASM recommendations [25]. Apnea was defined as the cessation of airflow for at least 10 s in the presence of respiratory effort, and hypopnea was identified as a > 30% reduction in airflow for at least 10 s and was associated with either a > 3% decrease in oxygen saturation or arousal. The apnea–hypopnea index (AHI) was calculated as the average number of apnea and hypopnea events per hour. Individuals with OSA were classified according to an AHI of > 5, whereas those with an AHI of < 5 were classified as primary snorers. The percentage of time spent in sleep with an oxygen saturation of < 90% were defined as T90.

### 2.4. Animal Model of Intermittent-Hypoxia Exposure and Cigarette-Smoke Exposure

Male wild type Wistar rats (6 weeks of age, 120–140 g body weight) were purchased from Beijing Vital River Laboratory Animal Technology Co., Ltd. (Beijing, China) and raised in a sterile animal facility. Our study was approved by the Ethical Committee of Tianjin Medical University General Hospital (IRB2021-DW-47).

All of the rats were divided randomly into four groups, with eight rats in each of the groups: an air-exposed group (NC), cigarette smoke-exposed group (CS), 5% concentration of intermittent hypoxia-exposed group (IH5%), and CS combined with 5% concentration of intermittent hypoxia-exposed group (CS + IH5%).The rats in the IH5% and CS + IH5% groups were placed in a homemade intermittent-hypoxia cabin [26] from 9:00 to 17:00 (sleep period). The cabin was intermittently filled with pure nitrogen for the 30 s (6 L/min per minute) to reach a minimum 5% oxygen concentration and then compressed air was added for 90 s (15 L/min per minute) to gradually increase the oxygen concentration to a maximum of 20.9%. Each cycle was 2 min. The oxygen concentration was measured by an oxygen-concentration monitor.

Rats in the CS group were exposed to active smoke from commercial cigarettes (Daqianmen cigarettes with filter purchased from Shanghai, China; tar 10 mg, nicotine 1 mg, CO 12 mg) in a homemade smoke-exposure device as described previously. The volume fraction of smoke in the smoke-exposure device was about 15% (*v*/*v*). Animals were exposed for 1 h twice-a-day from 7:30–8:30 and 17:30–18:30. The rats in the CS + IH5% group were exposed to intermittent hypoxia between two cigarette-smoke exposures on the same day. During the same period, the rats in the air group were in the homemade intermittent-hypoxia cabin, which was continuously charged with compressed air. After 12 weeks of exposure, the rats were sacrificed, and the lungs and spleens were prepared for microscopic observations (Pannoramic MIDI, 3DHISTECH, Budapest, Hungary) with H&E staining and transmission electron microscopy (Hitachi TEM system, Tokyo, Japan) [27].

### 2.5. Statistical Analysis

Results for variables that were normally distributed are provided as the means ± standard deviations. Results for variables that were not normally distributed are summarized as medians and compared using Mann–Whitney U-tests. Students’ *t*-tests were used in order to compare the means of two independent variables. A Spearman’s correlation analysis was used to evaluate the relationship between two variables. Differences were statistically evaluated by a one-way ANOVA followed by Fisher’s PLSD. *p* values of < 0.05 were considered significant. Error bars were used to indicate the SD. All of the statistical analyses used SPSS 20.0 (IBM, New York, NY, USA).

## 3. Results

### 3.1. General Clinical Characteristics of Patients Included

The flow-chart is shown in Figure 1. In the Entire OSA Cohort, we included 275 OSA patients. As some patients have more than one cause, there were 320 causes canvassed in total. The causes of chronic breathlessness in this group are shown in Table 1. Respiratory (159/275) and cardiac diseases (73/275) accounted for the vast majority of causes in the patients, followed by gastrointestinal and renal disorders. COPD and coronary artery disease were the most common lung and heart diseases, respectively. Other causes include obesity, age, and mood. Three patients had final diagnoses inconsistent with their initial department. One patient with suspected COPD was admitted to respiratory medicine, which finally confirmed motor neuron disease and referred the patient to neurology. Two patients were admitted to the cardiology department with suspected coronary heart disease. Their coronary angiographies were normal, and their final diagnoses were COPD.

The Final Study Population comprised 115 patients. The patients were all from the respiratory and cardiology departments. Among them, 51 were also diagnosed with OVS (39 men, 76.47%), with a mean age of 67.63 years (range 64.63–70.63 years), and 64 were diagnosed with OSA alone (44 men, 68.75%), with a mean age of 59.02 years (range 55.97–62.06 years). The control group included 13 healthy subjects (8 men, 61.54%) who underwent sleep monitoring as well as spirometry and were shown to have no OSA or OVS. In the OSA-alone group, 47 patients (73.4%) had small-airway dysfunction. In the OVS group, 12 patients had mild OSA, 14 had moderate OSA, and 25 had severe OSA. The proportion of moderate to severe OSA was 76.5%. The proportion of hypercapnia was 66.67%.

The OVS patients in the Final Study Population were all newly diagnosed (*n* = 3 for grade 1; *n* = 13 for grade 2; *n* = 19 for grade 3; *n* = 16 for grade 4). Surprisingly, 35 patients (68.6%) had an FEV1 of < 50%, indicating a very poor quality of life. The characteristics of the study population are shown in Table 2.

### 3.2. Lymphocytes and Tlymphocyte Subset Associated with Lung Ventilation

In the Final Study Population, a correlation analysis showed that lymphocyte parameters were related to FEV1 (r = 0.375, *p* < 0.001), FEV1% (r = 0.376, *p* < 0.001), FVC (r = 0.306, *p* = 0.001), FVC% (r = 0.347, *p* < 0.001), and FEV1/FVC (r = 0.277, *p* < 0.003). CD4 counts were related to FEV1 (r = 0.349, *p* < 0.001), FEV1% (r = 0.361, *p* < 0.001), FVC (r = 0.288, *p* = 0.002), FVC% (r = 0.323, *p* < 0.001), FEV1/FVC (r = 0.287, *p* = 0.002), and HCO_3_ (r = −0.189, *p* = 0.044) in OSA (Table 3).These results suggest that a decline in total lymphocyte and CD4 counts may be associated with poor alveolar hypoventilation. No correlation was found between CD3/CD45 counts or CD4/CD8 counts and the above-mentioned ventilation parameters. Different expressions of lymphocytes, CD4 counts, NLR, and PLR between the control, OSA-alone group, and OVS group are shown in Figure 2.

### 3.3. Value of Lymphocyte Parameters for Diagnosingovs in OSA

In the Final Study Population, the diagnostic values of lymphocyte parameters of OVS in OSA are shown in Figure 3. The ROC curve of the NLR had an AUC of 0.708 (95% CI, 0.614–0.802), with a cut-off of 2.246, a sensitivity of 0.843, and a specificity of 0.558. The ROC curve of the PLR had an AUC of 0.615 (95% CI, 0.508–0.722), with a cut-off of 172.06, a sensitivity of 0.451, and a specificity of 0.831. The ROC curve of lymphocytes had an AUC of 0.719 (95% CI, 0.624–0.813), with a cut-off of 1.79, a sensitivity of 0.578, and a specificity of 0.784. Whereas, the ROC curve of the CD4 counts had an AUC of 0.744 (95% CI, 0.653–0.834), with a cut-off of 542.846, a sensitivity of 0.625, and a specificity of 0.863.

### 3.4. Prognostic Value of Lymphocyte Parameters in OVS

All patients in the Final Study Population were followed up on for more than 6 months, and no patients died. Ten patients (19.6%) were re-admitted within 6 months, and 16 patients (31.37%) were re-admitted within 1 year, 5 of whom (9.8%) had two acute exacerbations within 1 year. All patients were hospitalized for COPD exacerbation. Additionally, 26 patients were treated with non-invasive ventilators during their hospital stay, and 25 patients (49.02%) wore non-invasive ventilators regularly outside of the hospital.

We assessed the NLR, CAT scores, PCO_2_, and CD4 counts for predicting a 6-month re-admission ROC curve. At the 6-month interval, the ROC curve of the NLR, CAT scores, PCO_2_,and CD4 counts had AUC values of 0.732 (95% CI, 0.584–0.879), 0.628 (95% CI, 0.477–0.779), 0.673 (95% CI, 0.475–0.871), and 0.678 (95% CI, 0.534–0.822), respectively (Figure 4). The NLR had the highest AUC for predicting 6-month re-admission, with a cut-off of 3.567 and a moderate prognostic value. The sensitivity and specificity were 0.8 and 0.732, respectively. The cut-off of CD4 counts for predicting 6-month re-admission was 392 cells/μL, indicating that patients with a CD4 count of less than 392 cells/μL had a poor prognosis.

We used binary logistic regression to evaluate the risk factors of 6-month re-admission. Comorbidities of cor pulmonale, coronary disease, diabetes mellitus, hypertension and smoking history, BMI, gender, age, total lymphocytes, CD3/CD45, CD4 counts, CD4 counts < 392 cells/μL, NLR, PLR, sleep monitoring data (AHI, SpO_2_min, mean SpO_2_, T90), PO_2_, PCO_2_, HCO_3_, CAT score, GOLD group, and whether a non-invasive ventilator was regularly used outside of the hospital were included in the univariate logistic regression analysis.CD4 counts of < 392 cells/μL (OR 12.706, CI 1.469, 109.887), cor pulmonale (OR 7.714, CI 1.44, 41.332), and age (OR 1.089, CI 1.004, 1.182) showed statistically significant differences of *p* < 0.05. These variables were used in the binary logistic regression analysis, which showed that CD4 counts of <392 cells/μL (aOR27.432, CI 2.018, 372.960), cor pulmonale (aOR16.702, CI 1.864, 149.673), and age (aOR1.111, CI 1.007, 1.227) were independent risk factors for 6-month re-admission (Table 4).

### 3.5. Spleen Damage in the Animal Model

Rats were randomly divided into four groups. After 16 weeks, the CS, IH5%, and CS + IH5% groups showed severe lung damage compared to the NC group (Figure 1). The CS + IH5% group had more damage and had more alveolar collapse and inflammatory cell infiltration than did the CS and IH5% groups. The body and spleen weights of the rats were significantly decreased in the CS, IH5%, and CS + IH5% groups (Figure 5A,B).

The spleen H&E staining showed that, compared to the NC group, the germinal center was decreased in the CS, IH5%, and CS + IH5% groups (Figure 5C). The CS + IH5% group had more germinal center damage than the CS and IH5% groups. This result was also confirmed by electron microscopy observation of the spleens. The cells in the spleens of the CS, IH5%, and CS + IH5% groups showed chromatin activation (Figure 6A) and mitochondrial swelling (Figure 6B) and was most severe in the CS + IH5% group, which showed that the spleen and immune cells suffered severe damage under the overlapping effect of intermittent hypoxia and cigarette smoke exposure.

## 4. Discussion

To our knowledge, this is the first reported study of chronic breathlessness in OSA patients. This is a highly select group, and there is little in the medical literature with which to compare our results. We further explored the role of lymphocyte parameters in the recognition of COPD in this population and their application value in prognostic judgments. Animal models reflected an immunosuppression of OVS. Besides excessive daytime sleepiness, OSA patients often presented with chronic breathlessness, which is a serious burden for their families and for the economy. Therefore, it is important to identify the cause and accurately triage patients to the appropriate department. Clinical scores are important for assessing OSA patients. In addition to the Epworth Sleepiness Scale (ESS) score, we believe that scores related to breathlessness should be added to routine evaluations of OSA patients.

There have been studies focused on the diagnostic value of the NLR and PLR between a control group and an OSA–COPD group [20]. Clinically, however, we pay more attention to patients who are prone to a misdiagnosis. Spirometry can be recommended as a “gold standard” for diagnosing COPD. However, the differences in testing levels and a lack of awareness while performing this test remain a global problem, especially in primary care [19]. Thus, a simple, inexpensive, and evidence-based approach to a chronic-breathlessness assessment that is accessible to primary care physicians may help to triage patients more efficiently and increase an awareness of the diagnosis. In our research, the Final Study Population included OSA patients with chronic breathlessness as the main complaint, which was a highly select group of patients who were prone to a misdiagnosis of COPD. We identified that the NLR had a moderate value for the recognition of COPD in this subset of patients, as well as for prognostic re-admission within 6 months. This group also included patients with OSA alone, of which a significant proportion showed small-airway pathology. Our research showed that the total lymphocyte and CD4 counts correlated with a lung-ventilation function. However, although the lymphocyte-related indicators we studied correlated with alveolar-ventilation function, the relevant analyses we performed did not reveal any indicators that reflected small-airway dysfunction early in the course of the disease.

A previous study showed that OVS patients were at an increased risk of recurrent episodes of acute exacerbation of COPD (RR 1.7 compared to patients with COPD) and recurrent acute hypercapnic respiratory failure [28]. Perhaps for such a population, non-invasive ventilators are beneficial, although the regular application of home non-invasive ventilators does not reduce readmissions within 6 months [29]. Further studies may be designed to observe the change in the CD4 counts in OVS during non-invasive ventilator therapy and further elucidate the correlation between CD4^+^T cells and ventilation function. There are many studies that focus on T lymphocytes and OSA. However, few studies focus on their effect on OVS and only describe the correlation of lymphocytes and OSA, lacking any prediction of the prognosis. Our study also demonstrated that CD4 counts can be used as an evaluation index for predicting OVS-patient readmission within 6 months. We concluded that a cutoff of 392 CD4^+^cells/μL was predictive, which may have practical value for clinical practice. Immunotherapy for OVS patients is also an area that requires more in-depth research.

Among the causes of chronic breathlessness of OSA caused by lung-related diseases, we found that interstitial disease was the most common cause, secondary only to COPD, which was consistent with the increasing attention in recent years of research on the high prevalence of interstitial disease combined with OSA. Asthma is not as common a cause as we expected, probably because most of these patients are treated on an outpatient basis and are rarely admitted to the hospital. Because our study was a retrospective study, in order to ensure the completeness of the patient data and the accuracy of the diagnosis, we only included hospitalized patients. This may create some selection bias, constituting one of the shortcomings of this study. Although lymphocyte parameters offered a moderately diagnostic and prognostic value, we believe these clinically accessible indicators may benefit some patients who are easily overlooked.

## Figures and Tables

**Figure 1 jcm-12-00936-f001:**
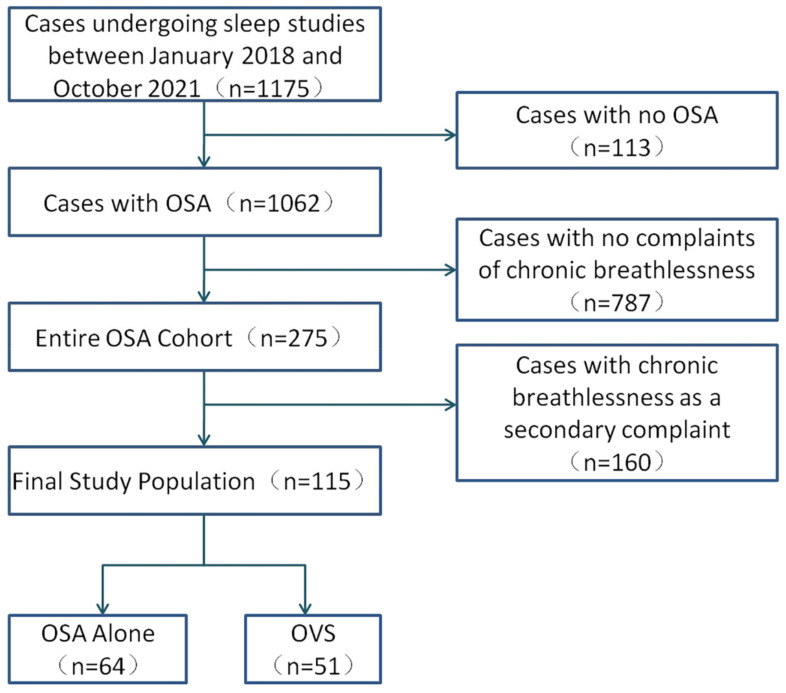
Flow-chart of the study cohort. Abbreviations: OSA, obstructive sleep apnea; OVS, Overlap syndrome.

**Figure 2 jcm-12-00936-f002:**
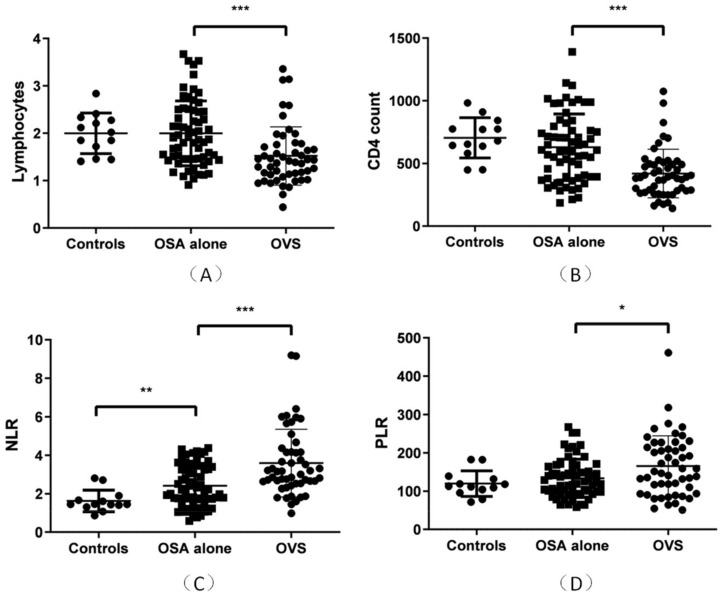
Expression of lymphocytes, CD4 counts, NLR, PLR in the OVS (*n* = 51), OSA-alone (*n* = 64), and healthy-control groups (*n* = 13). (**A**) Lymphocytes were significantly lower in the OVS group than in the OSA-alone group (*p* < 0.001) and were not significantly different between the healthy-control and OSA-alone groups. (**B**) CD4 counts were significantly lower in the OVS group than in the OSA-alone group (*p* < 0.001) and were not significantly different between the healthy-control and OSA-alone groups.(**C**) NLRs were significantly lower in the OSA-alone group than in the healthy-control group (*p* = 0.001) and was significantly lower in the OVS group than in the OSA-alone group (*p* < 0.001). (**D**) PLRs were significantly lower in the OVS group than in the OSA-alone group (*p* = 0.013) and was not significantly different between the control and OSA-alone groups. * *p* < 0.05, ** *p* < 0.01, *** *p* < 0.001.

**Figure 3 jcm-12-00936-f003:**
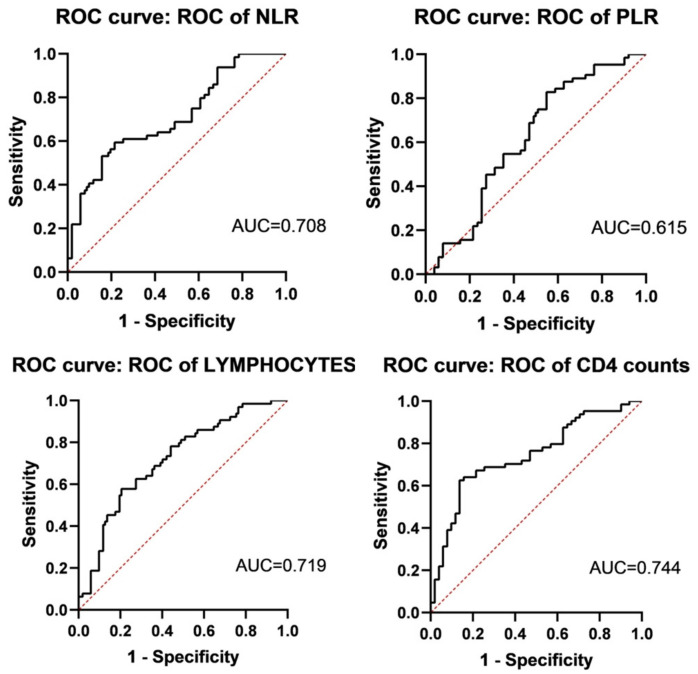
The ROC curve of lymphocyte parameters of diagnosing OVS (*n* = 51) in OSA (*n* = 115). ROC curve of NLR, PLR, lymphocytes, and CD4 counts were AUC = 0.708 (95% CI, 0.614–0.802), AUC = 0.615 (95% CI, 0.508–0.722), AUC = 0.719 (95% CI, 0.624–0.813), and AUC = 0.744 (95% CI, 0.653–0.834), respectively.

**Figure 4 jcm-12-00936-f004:**
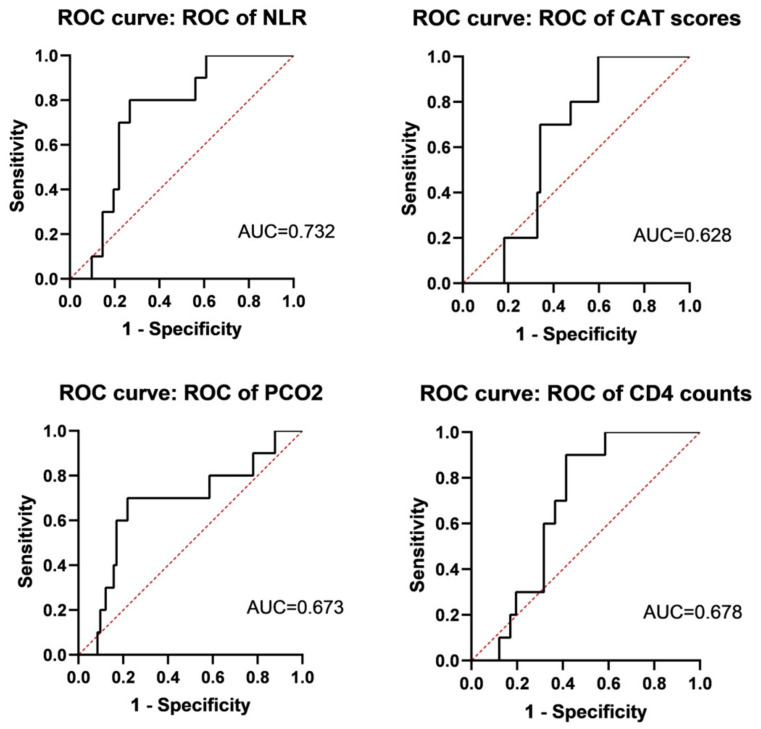
Evaluation of NLR, CAT scores, PCO_2_, and CD4 count for predicting 6-month readmission. ROC curves of NLR, CAT scores, PCO_2_, and CD4 count were AUC = 0.732 (95% CI, 0.584–0.879), AUC = 0.628 (95% CI, 0.477–0.779), AUC = 0.673 (95% CI, 0.475–0.871), and AUC = 0.678 (95% CI, 0.534–0.822), respectively.

**Figure 5 jcm-12-00936-f005:**
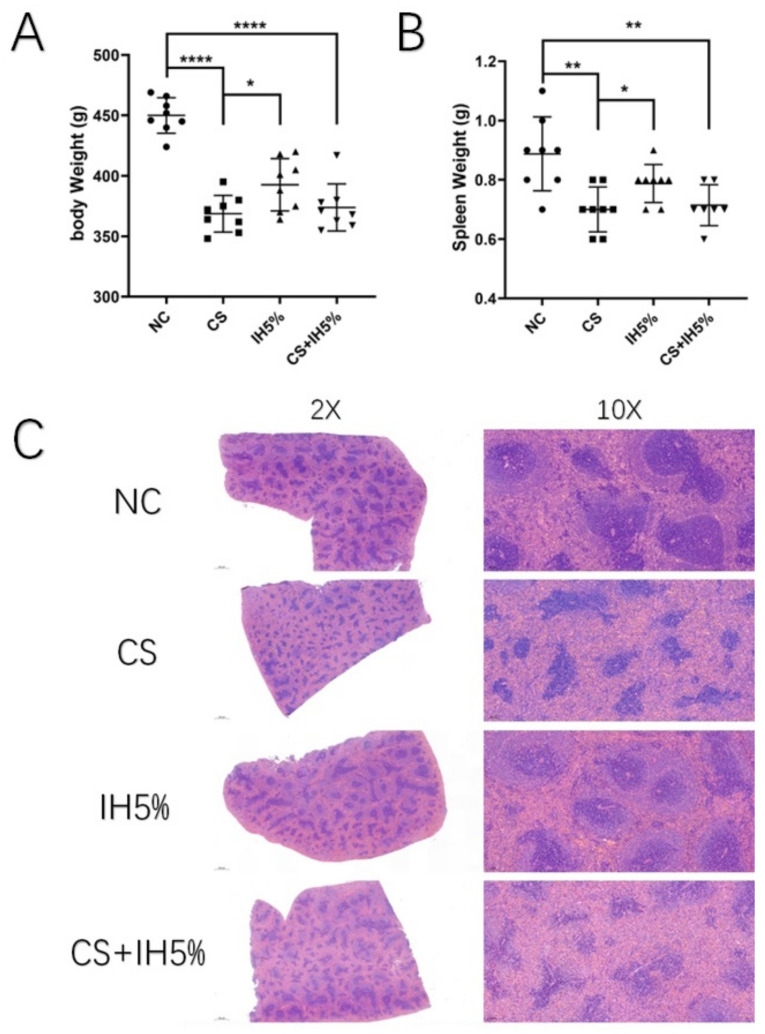
The rat model groups, NC, CS, IH5%, and CS + IH5%. (**A**) The body weight and (**B**) spleen weight of the NC, CS, IH5%, and CS + IH5% groups (*n* = 7–8; * *p* < 0.05; ** *p* < 0.01; **** *p* < 0.0001). (**C**) Representative H&E staining of the rat spleen.

**Figure 6 jcm-12-00936-f006:**
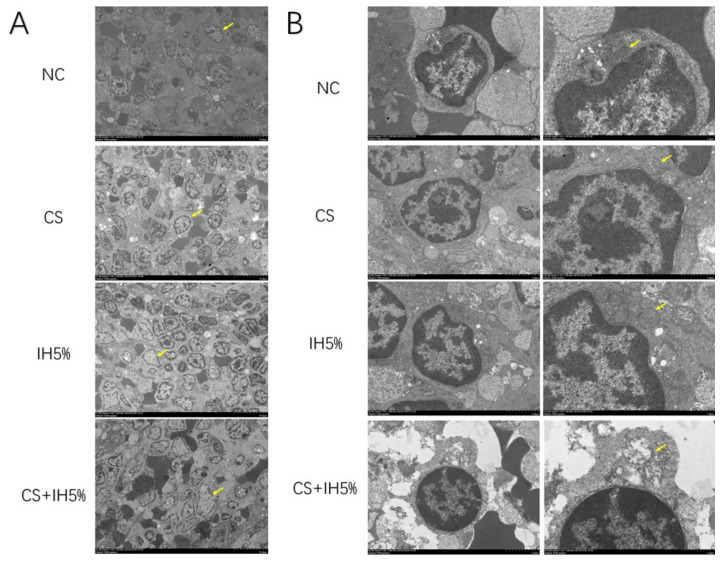
Representative transmission electron microscope observation of rat spleen. (**A**) ×1 k zoom (yellow arrows showed cells with chromatin activation). (**B**) ×5 k and ×10 k zoom (yellow arrows showed cells with chromatin activation).

**Table 1 jcm-12-00936-t001:** Cause of Chronic Breathlessness in 275 OSA Patients (Entire OSA Cohort).

Principal Organ System Involved (No. of Patients)	Diagnosis	No.of Patients
Respiratorytract diseases (159)	COPD	67
	ILD	51
	OHS	18
	Asthma	11
	Lung cancer	4
	Chronic lower respiratory bacterial infection	3
	Chronic pulmonary embolism	2
	Primary pulmonary hypertension	1
	Pneumoconiosis	1
	Pectus deformity	1
Cardiac disease (73)	Coronary artery disease	70
	Cardiomyopathy	2
	Arrhythmias	1
Gastrointestinal tract (32)	Gastroesophageal reflux	32
Kidney (18)	Chronic renal insufficiency	18
Neuromuscular diseases (7)	Stroke	5
	Guillain-Barré syndrome	1
	Motor neuron disease	1
Thyroid (6)	Thyrotoxicosis	6
Blood system (2)	Anaemia	2
Other causes (23)		23

Abbreviations: OSA, obstructive sleep apnea; OVS, Overlap syndrome; COPD, chronic obstructive pulmonary disease; OHS, obesity hypoventilation syndrome; ILD, interstitial lung disease.

**Table 2 jcm-12-00936-t002:** Clinical Characteristics of Final Study Population.

	OSA Alone (*n* = 64)	OVS (*n* = 51)	*p*-Value
Age (years)	59.02 (55.97, 62.06)	67.63 (64.63, 70.63)	<0.001
Male, *n* (%)	44 (68.75)	39 (76.47)	0.24
BMI (kg/m^2^)	31.76 (30.35, 33.18)	28.44 (27.08, 29.79)	0.001
Current and Former smokers, *n* (%)	29 (45.31)	41 (80.39)	<0.001
FEV1 (%predicted)	89.39 (84.95, 93.83)	42.33 (37.21, 47.45)	<0.001
FVC (%predicted)	91.35 (86.93, 95.76)	60.36 (55.58, 65.13)	<0.001
FEV1/FVC ratio	78.44 (75.49, 81.39)	54.94 (50.52, 59.36)	<0.001
Arterial blood gases			
PCO_2_ (mmHg)	42.27 (40.98, 43.55)	47.95 (45.40, 50.50)	<0.001
PO_2_ (mmHg)	72.04 (69.32, 74.75)	74.02 (69.64, 78.40)	0.424
HCO_3_ mmol/L	25.93 (25.23, 26.63)	28.32 (27.16, 29.47)	<0.001
Sleep Monitoring			
AHI (events/hr)	51.34 (44.41, 58.27)	33.79 (27.16, 40.43)	<0.001
Apnea (events/hr)	31.81 (22.54, 41.09)	10.11 (5.98, 14.23)	<0.001
ODI (events/hr)	46.49 (39.40, 53.57)	28.73 (23.12, 34.33)	<0.001
SPO_2_min (%)	69.98 (65.96, 74.01)	75.61 (72.47, 78.75)	0.036
Mean SpO_2_ (%)	91.69 (90.41, 92.96)	91.37 (90.43, 92.31)	0.704
T90(%)	21.45 (15.31, 27.59)	29.19 (20.25, 38.14)	0.143
WBC (×10^9^/L)	7.01 (6.60, 7.42)	7.11 (6.52, 7.69)	0.01
Neutrophils (×10^9^/L)	4.27 (3.94, 4.60)	4.85 (4.38, 5.32)	0.041
Thrombocytes (×10^9^/L)	244.02 (227.60, 260.63)	219.80 (201.73, 237.87)	0.05
Lymphocytes (×10^9^/L)	2.00 (1.83, 2.17)	1.52 (1.35, 1.69)	0.137
CD3+/CD45+	71.84 (70.22, 73.47)	69.02 (66.39, 71.66)	0.009
CD3+CD4+/CD45+	43.48 (41.16, 45.80)	40.30 (38.08, 42.52)	0.054
CD3+CD8+/CD45+	26.47 (24.11, 28.83)	27.56 (25.08, 30.05)	0.527
CD4/CD8	1.94 (1.70, 2.17)	1.62 (1.44, 1.80)	0.014
CD4 counts	629.03 (562.68, 695.37)	419.86 (365.33, 474.39)	0.008
NLR	2.41 (2.15, 2.68)	3.60 (3.10, 4.09)	<0.001
PLR	133.53 (121.06, 145.99)	165.59 (143.43, 187.75)	0.008
Comorbidities			
Cor pulmonale, *n* (%)	3 (4.69)	22 (43.14)	<0.001
Coronary disease, *n* (%)	17 (26.56)	26 (50.98)	0.006
Hypertension, *n* (%)	45 (70.31)	31 (60.78)	0.191
Diabetes mellitus, *n* (%)	15 (23.44)	9 (17.65)	0.3

Data are shown as mean ± SD or median [IQR]. Abbreviations: OSA, obstructive sleep apnea; OVS, Overlap syndrome; BMI, body mass index; FEV1,forced expiratory volume in 1 s; FVC, forced vital capacity; FEV1/FVC: ratio of FEV1 to FVC; AHI, apnea-hypopnea index; SpO_2_min, minimum peripheral capillary oxygen saturation; T90, total sleep time spent with oxygen saturation below 90%; ODI, oxygen desaturation index; WBC, white blood cells; NLR: neutrophil-to-lymphocyte ratio; PLR: platelet-to-lymphocyte ratio.

**Table 3 jcm-12-00936-t003:** Spearman’s correlation analysis between lymphocytes with a lung-ventilation function.

Parameters	FEV1	FEV1%	FVC	FVC%	FEV1/FVC	HCO_3_	PCO_2_
Lymphocytes	r = 0.375, *p* < 0.001	r = 0.376, *p* < 0.001	r = 0.306, *p* = 0.001	r = 0.347, *p* < 0.001	r = 0.277, *p* = 0.003	r = −0.148, *p* = 0.115	r = −0.178, *p* = 0.056
CD4 counts	r = 0.349, *p* < 0.001	r = 0.361, *p* < 0.001	r = 0.288, *p* = 0.002	r = 0.323, *p* < 0.001	r = 0.287, *p* = 0.002	r = −0.189, *p* = 0.044	r = −0.155, *p* = 0.099
NLR	r = −0.433, *p* < 0.001	r = −0.369, *p* < 0.001	r = −0.415, *p* <0.001	r = −0.380, *p* < 0.001	r = −0.207, *p* = 0.026	r = 0.221, *p* = 0.017	r = 0.252, *p* = 0.007
PLR	r = −0.182, *p* = 0.052	r = −0.153, *p* = 0.102	r = −0.168, *p* = 0.073	r = −0.122, *p* = 0.193	r = −0.188, *p* = 0.044	r = 0.047, *p* = 0.620	r = 0.056, *p* = 0.554

Data are shown as mean ± SD or median [IQR]. Abbreviations: FEV1, forced expiratory volume in 1 s; FVC, forced vital capacity; FEV1/FVC: ratio of FEV1 to FVC; NLR: neutrophil-to-lymphocyte ratio; PLR: platelet-to-lymphocyte ratio.

**Table 4 jcm-12-00936-t004:** Risk factors of 6-month readmission in OVS using a binary logistic regression analysis.

Parameters	Unadjusted Univariate		Multivariable	
	OR (95% CI)	*p*	OR (95% CI)	*p*
CD4 < 392 cell/uL	12.706 (1.469–109.887)	0.021	27.432 (2.018–372.960)	0.013
Cor pulmonale	7.714 (1.44–41.332)	0.017	16.702 (1.864–149.673)	0.012
Age	1.089 (1.004–1.182)	0.041	1.111 (1.007–1.227)	0.036

Data are shown as mean ± SD or median [IQR]. Abbreviations: OVS, overlap syndrome.

## Data Availability

The data presented in this study are available on request from the corresponding authors. The data are not publicly available due to privacy.

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
