# Peer review of "Chronic Breathlessness in Obstructive Sleep Apnea and the Use of Lymphocyte Parameters to Identify Overlap Syndrome among Patients"

_jcm, 2023, doi:10.3390/jcm12030936_

Round 1

Reviewer 1 Report

The present study was aimed to investigate overlap syndrome among consecutive adults who were diagnosed with OSA. Group1 included patients with complaint of chronic breathlessness who were hospitalized or a new diagnostic cause of their breathlessness. Group 2 included highly selected patients with chronic breathlessness as the primary complaint, not secondary as one of many other complaints. The article is interesting. However, there are several concerns:

1. There are several grammatical mistakes which needs to be addressed

2. Please highlight why Overlap syndrome needs to be detected and its significance in OSA patients

This is a very interesting study. The methodology was well written. However, the author needs to revise the abstract. Please highlight the aim and importance of this study. 

Author Response

Point 1: There are several grammatical mistakes which needs to be addressed.

Response 1: Thank you for your comment. We carefully reviewed the article and corrected the grammatical errors in it. The revisions have been marked up. Please review our manuscript again.

Point 2:Please highlight why Overlap syndrome needs to be detected and its significance in OSA patients.

Response 2:Thank you for your comment. We haverevised the part of the Abstract and highlighted the importance of detecting overlap syndrome in our study. Due to the word limit in the Abstract section, please check if our changes are appropriate.

Reviewer 2 Report

Dear authors,

These days, pulmonary function test is an available tool that can be used and interpreted by general physicians. Considering that spirometry is the gold standard for diagnosis of COPD, the value of the mentioned marker should be clarified in the manuscript.

Author Response

Point 1: These days, pulmonary function test is an available tool that can be used and interpreted by general physicians. Considering that spirometry is the gold standard for diagnosis of COPD, the value of the mentioned marker should be clarified in the manuscript.

Response 1: Thank you for your comment.Spirometry can be recommended as a “gold standard” to diagnose COPD. However, the differences in testing levels and lack of awareness of performing this test remain a global problem, especially in primary care. Thus, a simple, inexpensive, and evidence-basedapproach to chronic breathlessness assessment accessibleto primarycare physicians may help to triage patients more efficiently and increase awareness of the diagnosis.We have clarified the importance of mentioned marker in the Discussion section.

Reviewer 3 Report

Congratulations for the manuscript. It would be interesting to cite in the Introduction dental Authors who have dealt with this topic. It is important to cite works that have clinically treated this clinical problem from a dental and orthodontic point of view. You can find a lot about MDPI, Applied Science and special issues related to dentistry.

At line 64 ,before to start to describe other, thay could cite how the other clinicians treat OSA. They can found in MDPI , for exapmle, recent manuscripts about it (Dentist and Ortognatodontists) that improve the Introduction

The same , they can improve the Conclusion citing other kind of treatments because also they are important for the reader.

Author Response

Point 1: Congratulations for the manuscript. It would be interesting to cite in the Introduction dental Authors who have dealt with this topic. It is important to cite works that have clinically treated this clinical problem from a dental and orthodontic point of view. You can find a lot about MDPI, Applied Science and special issues related to dentistry.

At line 64 ,before to start to describe other, thay could cite how the other clinicians treat OSA. They can found in MDPI , for exapmle, recent manuscripts about it (Dentist and Ortognatodontists) that improve the Introduction

The same , they can improve the Conclusion citing other kind of treatments because also they are important for the reader.

Response 1: Thank you for your comment.Indeed, OSA is increasingly valued by multiple disciplines, not only in respiratory medicine. We cited three relevant articlesin the view of dental and orthodontics to improve our Introduction part.Please review our manuscript.